# Correlates of Calcidiol Deficiency in Adults—Cross-Sectional, Observational, Population-Based Study

**DOI:** 10.3390/nu14030459

**Published:** 2022-01-20

**Authors:** Massimo Cirillo, Giancarlo Bilancio, Pierpaolo Cavallo, Simona Costanzo, Amalia De Curtis, Augusto Di Castelnuovo, Licia Iacoviello

**Affiliations:** 1Department of Public Health, University of Naples “Federico II”, 80131 Naples, Italy; 2Department of Medicine, Surgery and Odontoiatry, “Scuola Medica Salernitana”, University of Salerno, 84081 Baronissi, Italy; giancarlo.bilancio@gmail.com; 3Department of Physics, University of Salerno, 84084 Fisciano, Italy; pcavallo@unisa.it; 4Department of Epidemiology and Prevention, Istituto di Ricovero e Cura a Carattere Scientifico (IRCCS) Neuromed, 86077 Pozzilli, Italy; simona.costanzo@moli-sani.org (S.C.); amalia.decurtis@moli-sani.org (A.D.C.); licia.iacoviello@moli-sani.org (L.I.); 5Mediterranea Cardiocentro, 80122 Naples, Italy; dicastel@ngi.it; 6Department of Medicine and Surgery, Research Center in Epidemiology and Preventive Medicine (EPIMED), University of Insubria, 21100 Varese, Italy

**Keywords:** 25-hydroxyvitamin D (calcidiol), physical activity, abdominal obesity, cholesterol, smoking, alcohol

## Abstract

The prevalence, determinants, and clinical significance of vitamin D deficiency in the population are debated. The population-based study investigated the cross-sectional associations of several variables with serum 25-hydroxyvitamin D (calcidiol) measured using standardized calibrators. The study cohort consisted of 979 persons of the Moli-sani study, both sexes, ages ≥35 years. The correlates in the analyses were sex, age, education, local solar irradiance in the month preceding the visit, physical activity, anthropometry, diabetes, kidney function, albuminuria, blood pressure, serum cholesterol, smoking, alcohol intake, calorie intake, dietary vitamin D intake, and vitamin D supplement. The serum calcidiol was log transformed for linear regression because it was positively skewed (skewness = 1.16). The prevalence of calcidiol deficiency defined as serum calcidiol ≤12 ng/mL was 24.5%. In multi-variable regression, older age, lower solar irradiance, lower leisure physical activity, higher waist/hip ratio, higher systolic pressure, higher serum cholesterol, smoking, lower alcohol intake, and no vitamin D supplement were independent correlates of lower serum calcidiol (95% confidence interval of standardized regression coefficient ≠ 0) and of calcidiol deficiency (95% confidence interval of odds ratio > 1). The data indicate that low serum calcidiol in the population could reflect not only sun exposure, age, and vitamin D supplementation but also leisure physical activity, abdominal obesity, systolic hypertension, hypercholesterolemia, smoking, and alcohol intake.

## 1. Introduction

Calcitriol, also named 1,25-dihydroxyvitamin D, is regarded as the most active vitamin D form [1]. The generation of calcitriol is quite complex because it includes at least four different steps: the endogenous synthesis of cholecalciferol in the skin after ultraviolet exposure (vitamin D_3_); the absorption of dietary ergocalciferol (vitamin D_2_); the conversion of vitamin D_3_ and D_2_ to 25-hydroxyvitamin D (calcidiol) through a hydroxylation at C-25 due to the activity of liver cytochrome CYP2R1 and perhaps other cytochromes; and the final conversion of calcidiol to calcitriol due to a hydroxylation at C-1 by 1α-hydroxylase, which is present mainly but not solely in the kidney [1]. Some authors reported that the prevalence of vitamin D deficiency is high in the population based on the evidence of serum calcidiol below 20 ng/mL [2,3,4]. Others argued that this conclusion misinterpreted the concept of vitamin D deficiency because serum calcidiol < 20 ng/mL rarely implies true vitamin D deficiency [5,6]. Regarding calcidiol deficiency, two consensus conferences recently underlined that the use of a calcidiol assay non-standardized with specific calibrators reduces the reliability of calcidiol measurements and that the threshold of 12 ng/mL (i.e., 30 nmol/L) should be considered for the definition of the risk of rickets/osteomalacia [7,8]. Regarding epidemiological studies on calcidiol deficiency [9,10,11,12], only one was based on the use of a standardized calcidiol assay [12]. Therefore, the aim of the present study was to investigate serum calcidiol in a sample of the Italian general population with the use of a standardized calcidiol assay and with the focus on possible correlates or determinants of calcidiol deficiency.

## 2. Materials and Methods

### 2.1. Study Design and Population Sample

The Moli-sani study is an ongoing cohort study that enrolled 24,325 individuals from 2005 to 2010, men and women, age 35 and over, randomly recruited from the general population of Molise, a region of central-southern Italy [13]. The study complies with the Declaration of Helsinki of 1975, as revised in 2013, and was approved by the Rome Catholic University ethical committee (P99, A.931/03-138-04, 11 February 2004). All participants provided written informed consent. The baseline visit was conducted at the Research Laboratories of the Catholic University in Campobasso (Italy) and included the following: three measurements of blood pressure and heart rate in the non-dominant arm by an automatic device (OMRON-HEM-705CP, Omron, Kyoto, Japan) with participants lying down for 5 min [13]; measurements of weight and height; the administration of the validated Italian food frequency questionnaire of the European Prospective Investigation into Cancer and Nutrition (EPIC) with assessment of habitual intakes in the past year of energy, macronutrients, micronutrients, and vitamins [14]; the administration of specific supplementary questions on alcohol intake of the Italian EPIC questionnaire [15]; questionnaires about education, habitual physical activity, smoking, dietary or pharmacological treatment(s), including vitamin supplements; the collection of untimed urine spot samples from the first void at wake up and of morning venous blood samples after an overnight fast. Biological samples were processed for lab tests within 3 h and/or stored in liquid nitrogen as described [16]. Lab tests for the whole cohort included the measurements of serum levels of glucose, total cholesterol, high-density lipoprotein cholesterol (HDL-cholesterol), and cystatin C as part of the BiomarCaRE project [17]. 

Target cohort for the present analysis consisted of 1000 examinees of the Moli-sani study that were selected by a sex- and age-stratified randomization for additional data collection [18]. As shown in Appendix A, the stratification was designed to have 100 men and 100 women for each one of the following five age-groups: 35–44, 45–54, 55–64, 65–74, and ≥75 years. The additional data collection in this target cohort included the average local solar irradiance in the month preceding the blood withdrawal as objective index of ultraviolet exposure and the measurements by automated biochemistry of serum calcidiol, serum creatinine, urine albumin, urine total protein, and urine markers of diet as reported [18,19]. Serum calcidiol was measured by a chemiluminescent assay (Diasorin, Saluggia, Italy) calibrated with ID-LC-MS- and ID-LC-MS/MS-traceable standard NIST-SRM 972a as per guidelines [7,8]. The prevalence of low serum calcidiol in the study cohort varied with the use of different assays and decreased with the use of NIST SRM 972a [18]. Serum creatinine was measured by an enzymatic assay calibrated with IDMS-traceable standard [20]. Intra- and inter- assay variability of all lab measurements was <5%.

### 2.2. Variables under Study

The study investigated the cross-sectional relations with serum calcidiol and with calcidiol deficiency of the following independent variables: sex, age, education, average local solar irradiance in the month preceding the blood withdrawal (from here on defined as solar irradiance), habitual physical activity in leisure time, anthropometry, blood pressure, serum cholesterol, diabetes, kidney function, albuminuria, smoking, habitual alcohol intake, habitual dietary intake of calorie, habitual dietary intake of vitamin D, and regular use of vitamin D supplements. Calcidiol deficiency was defined as serum calcidiol ≤12 ng/mL [7,8]. Data were given also for serum calcidiol <20 ng/mL for comparability to the single epidemiological study based on standardized serum calcidiol assay [12]. High education was defined as the report of high school diploma or higher and was used as proxy of the socio-economic status. Solar irradiance was used as objective index of ultraviolet exposure, which is a key determinant of skin vitamin D_3_ synthesis [21]. Solar irradiance was derived from the Italian Atlas of Solar Irradiation Database of the Italian National Agency for New Technologies, Energy and Sustainable Economic Development and expressed as daily megajoule per square meter (MJ/m^2^ per day) [22]. Regarding anthropometry, 24-h urinary creatinine was estimated using the Chronic Kidney Disease (CKD) Epidemiology Collaboration equation and used as index of muscle mass [23,24]; body mass index was calculated as weight_kg_/height_m_^2^ and used as index of overweight; waist/hip ratio was used as index of abdominal obesity and defined as high when ≥1 in men and ≥0.86 in women [25]. Questionnaire data were used for habitual physical activity in leisure time, which was expressed as metabolic equivalent of task per day (MET-h/day). Kidney function was assessed as estimated glomerular filtration rate (eGFR), which was calculated by the combined creatinine-cystatin C equation of the Chronic Kidney Disease Epidemiology Collaboration study to reduce the confounding of creatinine generation from skeletal muscle mass [26,27]. Urinary albumin was assessed as urinary albumin/creatinine ratio [28]. Hypercholesterolemia was defined as serum total cholesterol ≥240 mg/dL; non-HDL-cholesterol was calculated as the difference between serum total cholesterol and serum HDL-cholesterol. Smoking status and habitual alcohol intake were defined using questionnaire data [13,14,15]. Diabetes was defined as the report of regular treatment with anti-diabetic drug(s) or serum glucose ≥126 mg/dL. Data of the EPIC-food frequency questionnaire were used to assess the habitual intake of calories as kcal/day, vitamin D as IU/day, and alcohol intake as g/day [14,15]. Questionnaire data were used for definition of regular use of vitamin D supplements.

### 2.3. Statistics

The first set of analyses included single- and multi-variable linear regression to investigate the relation of correlates to serum calcidiol. Skewed variables were logarithm transformed in regression. For direct comparability among correlates, the results were reported as standardized regression coefficient (beta), that is, as the fraction of the standard deviation of the dependent variable (serum calcidiol) explained by a difference of one standard deviation in the given independent variable. An additional multi-variable linear regression model was analyzed to assess if findings differed between serum HDL-cholesterol and non-HDL cholesterol. The second set of analyses included single- and multi- variable logistic regression to investigate the correlates of calcidiol deficiency. To focus on calcidiol deficiency, regression coefficients were expressed for one lower standard deviation (SD) for correlates positively related to serum calcidiol and for one higher SD for correlates negatively related to serum calcidiol. Regression coefficients of categorical variables were expressed for difference between categories. Logistic coefficients were exponentiated for reporting the results as odds ratio. Based on the results of multi-variable logistic regression, the third set of analyses was designed to focus on the subgroup of the significant correlates of calcidiol deficiency that could be considered possible cofactors in the development of calcidiol deficiency. Two criteria were used to select these correlates: being considered amenable of control and non-affected by vitamin D status. The analyses excluded sex, age, and solar irradiance for being considered non-controllable and blood pressure status for being considered likely affected by vitamin D status [29]. This last set of analyses was limited to examinees not reporting the use of vitamin D supplement and included chi-square analysis and area under the receiver operating characteristic curve (ROC curve). All results were reported including 95% confidence interval (95% CI). Statistical procedures were performed using IBM-SPSS Statistics 19 (IBM, Armonk, NY, USA).

## 3. Results

### 3.1. Descriptive Statistics

The study cohort consisted of 979 examinees with complete data for serum calcidiol (mean ± SD, ng/mL = 21.6 ± 12.3 in the whole study cohort, 22.1 ± 11.9 in men, and 21.0 ± 12.7 in women). The serum calcidiol was log transformed for linear regression (mean ± SD = 1.26 ± 0.25 log ng/mL) because it was positively skewed (skewness ± SE = 1.16 ± 0.08, Appendix A). The prevalence was 24.5% for calcidiol deficiency (serum calcidiol <12 ng/mL) and 53.2% for serum calcidiol <20 ng/mL. The prevalence was higher in older ages both for calcidiol deficiency and serum calcidiol <20 ng/mL (Appendix A). Table 1 reports the descriptive statistics for the correlates in analysis. 

Leisure physical activity, urinary albumin/creatinine ratio, habitual alcohol intake, and habitual dietary intake of vitamin D were log transformed in regression analyses because they were positively skewed (skewness > 1). Due to the presence of zero values, leisure physical activity and alcohol intake were log-transformed after adding a negligible value to all the values (0.01 MET-h/day for physical activity and 0.01 g/day for alcohol intake, respectively). The habitual dietary intake of vitamin D ranged from 8 to 339 IU/day (min to max).

### 3.2. Linear Regression

Table 2 shows beta values with 95% CI in single- and multi- variable regression with log serum calcidiol as the dependent variable. In the single-variable analyses, beta was significantly negative (95% CI < 0) for female sex, age, body mass index, waist/hip ratio, diabetes, urinary albumin/creatinine ratio, systolic pressure, serum cholesterol, and smoking. Beta was significantly positive (95% CI > 0) for solar irradiance, leisure physical activity, alcohol intake, and vitamin D supplement. Beta was not significant for other variables (95% CI including zero). In the multi-variable analyses, beta was independently negative for age, waist/hip ratio, eGFR, systolic pressure, serum total cholesterol, and smoking, whilst it was independently positive for solar irradiance, leisure physical activity, alcohol intake, and vitamin D supplement. The R^2^ value of the model in Table 2 was 0.205. In an additional multi-variable model including serum HDL-cholesterol and non-HDL-cholesterol in the place of serum total cholesterol, beta was significant for non-HDL cholesterol (95% CI = −0.157/−0.039), non-significant for serum HDL-cholesterol (95% CI = −0.126/0.006), and almost identical to the values in Table 2 for other variables (not shown). 

### 3.3. Logistic Regression

Table 3 shows the results of single- and multi- variable logistic regression with calcidiol deficiency as the dependent variable. In simple regression, the odds ratio of calcidiol deficiency was significantly increased (95% CI > 1) for female sex, older age, lower solar irradiance, lower leisure physical activity, higher urinary creatinine, higher urinary albumin/creatinine ratio, higher blood pressure, higher serum total cholesterol, lower alcohol intake, higher calorie intake, lower vitamin D intake, and lack of vitamin D supplementation. The odds ratio was not significantly different from 1 for other variables. In multi-variable regression, the odds ratio of calcidiol deficiency was independently increased for older age, lower solar irradiance, lower leisure physical activity, higher waist/hip ratio, higher systolic pressure, higher serum total cholesterol, smoking, lower alcohol intake, and no vitamin D supplementation. In the multi-variable logistic model with serum calcidiol <20 ng/mL as the dependent variable, the findings were similar for solar irradiance, leisure physical activity, waist/hip ratio systolic pressure, serum total cholesterol, alcohol intake, and vitamin D supplements (Appendix A).

### 3.4. Controllable Correlates of Calcidiol Deficiency

Based on the results of the multi-variable logistic regression, the analyses on the controllable correlates of calcidiol deficiency focused on the following five traits: no leisure physical activity, high waist/hip ratio, hypercholesterolemia, smoking, and no alcohol intake. These five traits were defined as categorical variables (yes/no = 1/0). In the 952 examinees not reporting the use of vitamin D supplements, the prevalence of calcidiol deficiency was higher in the presence of any one of these traits and was linearly higher with increasing the number of traits (Figure 1).

For the detection of calcidiol deficiency, the area under the ROC curve of the number of traits was 0.638 (95% CI = 0.997/0.679, *p* < 0.001). The area was greater but not significantly in men compared to women (0.620 and 0.606, 95% CI = 0.555/0.685 and 0.549/0.663) and in persons ≥65 years of age compared to persons < 65 years of age (0.693 and 0.628, 95% CI = 0.612/0.775 and 0.581/0.675).

## 4. Discussion

The present study in a sample of the Italian adult population showed three main findings: (i) the dietary vitamin D intake ranged below the recommended daily allowance [30]; (ii) the dietary vitamin D intake did not relate to the serum calcidiol concentration or to the prevalence of calcidiol deficiency; (iii) independent associations with lower serum calcidiol or with higher prevalence of calcidiol deficiency were found for lower solar irradiance, lower physical activity in leisure time, higher waist/hip ratio, higher serum cholesterol, smoking, lower alcohol intake, and lack of vitamin D supplementation. 

The study limitations were the sample size, the lack of data for various ethnic groups and for ages <35 years, and the lack of information on genetic factors and personal habits of sun exposure. The study merits were the use of standardized calibrators for calcidiol measurements [7,8], the data collection for several possible correlates of serum calcidiol, including dietary vitamin D and an objective index of local solar irradiance, and the accurate calculation of eGFR with the use of serum concentrations of both creatinine and cystatin C [26].

The low range of dietary vitamin D intake in the study cohort and lack of association of vitamin D intake with serum calcidiol or calcidiol deficiency suggested that factors other than dietary vitamin D are more important determinants of serum calcidiol when dietary vitamin D intake is low. Ultraviolet-induced endogenous synthesis of vitamin D and use of vitamin D supplements appeared as the most powerful determinants of serum calcidiol levels because they had the highest beta in multiple-variable linear regression targeting serum calcidiol. In addition to sun exposure and vitamin D supplementation, the study results indicated a role in calcidiol deficiency for lower leisure physical activity, abdominal obesity, higher serum cholesterol, smoking, and lower alcohol intake, which independently related to both serum calcidiol and calcidiol deficiency. Approximately 80% of the cases with calcidiol deficiency were found in persons who had no leisure physical activity, high waist/hip ratio, hypercholesterolemia, were smokers, or had no alcohol intake. Moreover, the prevalence of calcidiol deficiency increased progressively with increasing the cumulative prevalence of these traits after control for sex, age, and solar irradiance.

Cross-sectional associations should be interpreted cautiously regarding the possible underlying mechanism(s). Habitual physical activity in leisure time could stimulate skin vitamin D synthesis due to an up-regulation of cutaneous blood flow [31] and/or via trophic effects on the skeletal muscle mass, which is capable of extending the calcidiol half-life [32]. Abdominal obesity could lower serum calcidiol, increasing the sequestration and/or the catabolism of calcidiol in the adipose tissue [33]. The lack of independent associations for body mass index in the multi-variable model, including waist/hip ratio, suggested that abdominal fat could be more important that non-abdominal fat for the unfavorable effects on calcidiol levels. Smoking could lower serum calcidiol, reducing the vitamin D skin generation via unfavorable effects on cutaneous blood flow [34] or on skin aging [35,36]. The association of habitual alcohol intake with serum calcidiol should be explained by long-term effects of alcohol intake given that Mahabir et al. reported that the short-term administration of alcohol did not modify serum calcidiol [37]. Theoretically, these long-term effects could include an induction of hepatic cytochrome hydroxylases [38] that are likely involved in the transformation of non-hydroxylated vitamin D into calcidiol [1]. The pathway of hepatic hydroxylation could also underly the association of hypercholesterolemia with serum calcidiol given that a diet-induced increase in serum cholesterol lowers both hepatic vitamin D-25-hydroxylase expression and serum calcidiol in an animal model [39]. Last, the association of higher eGFR with lower serum calcidiol could suggest that higher hydroxylation by renal 1α-hydroxylase could shorten the calcidiol half-life.

In comparison to previous studies, the present results were in accordance with the data of the EPIC study for the levels of dietary intake of vitamin D in other population samples [40], with several epidemiological studies on the prevalence of low serum calcidiol [30], with the effects reported for sun exposure [41,42], and with epidemiological data for the associations with calcidiol deficiency of physical activity, smoking, use of vitamin D supplements, or alcohol intake [11,12,43,44]. In support of the favorable effects of alcohol intake on serum calcidiol, there is also the observation that dietary patterns, including consumption of wine, were associated with higher bone mineral density [45]. Contrarily, the present results were at variance with the association of dietary vitamin D intake with serum calcidiol found in 596 community-dwelling Dutch elderly [42] and with the lack of association of serum cholesterol with serum calcidiol in 295 men from the New Zealand general population [43]. To the authors’ knowledge, this is the first report of an association between serum cholesterol and calcidiol in humans.

If calcidiol deficiency per se had a true clinical importance, the practical implications of the present results would be that it should be searched more actively not only in persons with low sun exposure but also in smokers, non-drinkers, persons with abdominal obesity, or hypercholesterolemia. Conversely, if calcidiol deficiency per se had limited or no clinical importance, the present results would suggest that the association of calcidiol deficiency with several diseases could possibly be due to its association with unfavorable traits, such as sedentarism, abdominal obesity, hypercholesterolemia, and smoking.

## 5. Conclusions

The study showed that, in a sample of the general population residing in southeastern Italy, vitamin D dietary intake was well below the recommended allowance, and the independent predictors of calcidiol deficiency were not only low solar irradiance but also factors amenable of control, such as sedentarism, abdominal obesity, smoking, and hypercholesterolemia. Therefore, the study results indicate that differences in the prevalence of calcidiol deficiency among different study cohorts may reflect differences in the prevalence of these factors. For patients, the study results imply that the prevention and control of calcidiol deficiency could be favored by the control of these factors as well.

## Figures and Tables

**Figure 1 nutrients-14-00459-f001:**
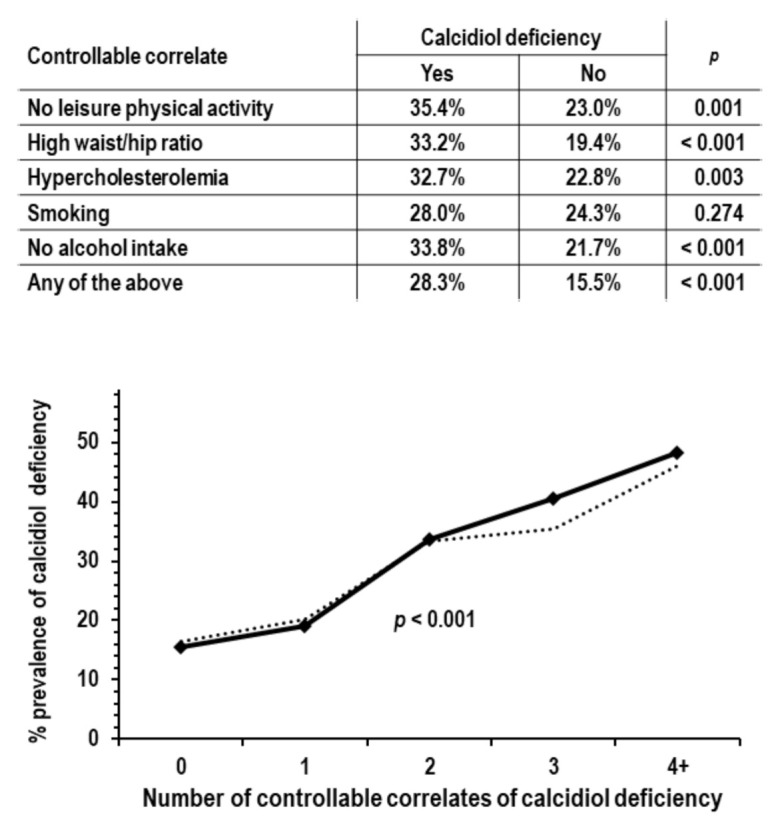
Analyses on the associations of controllable traits correlated with calcidiol deficiency; analyses were limited to the 952 examinees not reporting the use of vitamin D supplements; *p* values are by chi-square analysis. *Upper tabular insert*: calcidiol deficiency prevalence in the group with no leisure physical activity compared to the group with leisure physical activity (*n* = 158 and 794), in the group with high waist/hip ratio compared to the group without high waist/hip ratio (*n* = 394 and 558), in the group with hypercholesterolemia compared to the group without hypercholesterolemia (*n* = 223 and 729), in smokers compared to non-smokers (*n* = 207 and 745), in non-drinkers compared to drinkers (*n* = 269 and 683), and in the group with any of the above traits compared to the group without any of the above traits (*n* = 713 and 239). *Lower graph*: prevalence of calcidiol deficiency by number of controllable traits correlated with calcidiol deficiency (range from 0 to 5, *n* = 239, 336, 247, 101, 27, and 2). Due to low *n*, groups with 4 and 5 traits were combined in a single group indicated as 4+. The dotted line indicates prevalence of calcidiol deficiency in ANOVA with control for sex, age, solar irradiance, systolic pressure, and eGFR.

**Table 1 nutrients-14-00459-t001:** Descriptive statistics: prevalence for categorical variables, median (IQR) for skewed variables (skewness > 1), and mean ± SD for non-skewed variables and log-transformed skewed variables.

Sex, men/women = 0/1	501/478
Age, years	59.9 ± 9.8
High education, no/yes = 0/1	561/418
Daily solar irradiance, MJ/m^2^	14.5 ± 6.8
Leisure physical activity, MET-h/day log MET-h/day	2.27 (0.73/4.91) 0.029 ± 0.995
Urinary creatinine, g/24-h	1.27 ± 0.32
Body mass index, kg/m^2^	28.7 ± 4.9
Waist/hip ratio	0.927 ± 0.075
Diabetes, no/yes = 0/1	124/855
eGFR, mL/min × 1.73 m^2^	83 ± 16
Urinary albumin/creatinine ratio, mg/g log mg/g	8.8 (3.9/20.5) 0.95 ± 0.54
Systolic pressure, mm Hg	146 ± 20
Diastolic pressure, mm Hg	83 ± 9
Serum total cholesterol, mg/dL	213 ± 40
Serum HDL cholesterol, mg/dL	57 ± 14
Serum non-HDL cholesterol, mg/dL	156 ± 38
Smoking, no/yes = 0/1	210/769
Alcohol intake, g/day log g/day	8.7 (0.0/27.2) 0.55 ± 1.08
Calorie intake, kcal/day	2062 ± 664
Dietary vitamin D, IU/day log IU/day	83.8 (62.6/110.1) 1.91 ± 0.20
Vitamin D supplement, no/yes = 0/1	952/27

eGFR = estimated glomerular filtration rate. HDL = high-density lipoprotein.

**Table 2 nutrients-14-00459-t002:** Single-variable and multi-variable standardized regression coefficient (beta) to log serum calcidiol as dependent variable.

	Beta (95% CI)	
	Single-Variable Regression	Multi-Variable Regression
Sex, men/women = 0/1	**−0.069 (−0.132/−0.006)**	−0.013 (−0.239/0.213)
Age, years	**−0.115 (−0.178/−0.052)**	**−0.121 (−0.229/−0.013)**
High education, no/yes = 0/1	0.050 (−0.013/0.113)	−0.003 (−0.066/0.059)
Daily solar irradiance, MJ/m^2^	**0.255 (0.194/0.316)**	**0.229 (0.170/0.287)**
Leisure physical activity, log MET-h/day	**0.158 (0.095/0.0.221)**	**0.115 ((0.054/0.177)**
Urinary creatinine, g/24-h	0.046 (−0.017/0.109)	0.123 (−0.136/0.381)
Body mass index, kg/m^2^	**−0.112 (−0.175/−0.049)**	0.093 (−0.232/0.045)
Waist/hip ratio	**−0.136 (−0.199/−0.073)**	**−0.126 (−0.193/−0.059)**
Diabetes, no/yes = 0/1	**−0.075 (−0.138/−0.012)**	−0.012 (−0.074/0.049)
eGFR, mL/min × 1.73 m^2^	−0.008 (−0.071/0.055)	**−0.129 (−0.204/−0.054)**
Urinary albumin/creatinine ratio, log mg/g	**−0.102 (−0.165/−0.039)**	−0.041 (−0.101/0.020)
Systolic pressure, mm Hg	**−0.142 (−0.205/−0.079)**	**−0.120 (−0.207/−0.034)**
Diastolic pressure, mm Hg	−0.038 (−0.101/0.025)	0.047 (−0.034/0.128)
Serum total cholesterol, mg/dL	**−0.170 (−0.233/−0.107)**	**−0.110 (−0.169/−0.050)**
Serum HDL cholesterol, mg/dL	**−0.073 (−0.136/−0.010)**	not included
Serum non-HDL cholesterol, mg/dL	**−0.150 (−0.213/−0.087)**	not included
Smoking, no/yes = 0/1	**−0.066 (−0.129/−0.003)**	**−0.092 (−0.150/−0.033)**
Alcohol intake, log g/day	**0.071 (0.008/0.134)**	**0.075 (0.008/0.142)**
Calorie intake, kcal/day	0.029 (−0.034/0.092)	−0.043 (−0.120/0.034)
Dietary vitamin D, log IU/day	0.040 (−0.023/0.103)	0.027 (−0.043/0.097)
Vitamin D supplement, no/yes = 0/1	**0.150 (0.087/0.213)**	**0.200 (0.142/0.258)**

eGFR = estimated glomerular filtration rate. HDL = high-density lipoprotein. **Bold character** for statistically significant beta (95% CI not including zero).

**Table 3 nutrients-14-00459-t003:** Odds ratio (95% CI) of calcidiol deficiency in single-variable and multiple-variable logistic regression.

Independent Variables	Reference Interval	Odds Ratio (95% CI) of Calcidiol Deficiency	
		Single-Variable Regression	Multiple-Variable Regression
Sex	women vs. men	**1.78 (1.32/2.39)**	1.35 (0.38/4.77)
Age, years	+1SD	**1.32 (1.13/1.53)**	**1.37 (1.01/1.87)**
High education	No vs. yes	1.33 (0.99/1.79)	1.04 (0.73/1.47)
Daily solar irradiance, MJ/m^2^	−1SD	**1.53 (1.31/1.78)**	**1.53 (1.30/1.81)**
Leisure physical activity, log MET-h/day	−1SD	**1.36 (1.18/1.56)**	**1.24 (1.06/1.46)**
Urinary creatinine, g/24-h	+1SD	**1.30 (1.12/1.51)**	1.20 (0.58/2.50)
Body mass index, kg/m^2^	+1SD	1.03 (0.99/1.06)	1.12 (0.76/1.64)
Waist/hip ratio	+1SD	1.12 (0.97/1.30)	**1.21 (1.12/1.44)**
Diabetes	Yes vs. no	1.49 (0.99/2.25)	1.17 (0.72/1.90)
eGFR, mL/min × 1.73 m^2^	+1SD	0.95 (0.82/1.10)	1.23 (0.99/1.52)
Urinary albumin/creatinine ratio, log mg/g	+1SD	**1.24 (1.07/1.44)**	1.08 (0.92/1.28)
Systolic pressure, mm Hg	+1SD	**1.25 (1.08/1.44)**	**1.29 (1.03/1.63)**
Diastolic pressure, mm Hg	+1SD	**1.11 (1.04/1.18)**	0.88 (0.72/1.09)
Serum total cholesterol, mg/dL	+1SD	**1.33 (1.14/1.54)**	**1.24 (1.05/1.46)**
Smoking	Yes vs. no	1.23 (0.87/1.73)	**1.48 (1.01/2.19)**
Alcohol intake, log g/day	−1SD	**1.23 (1.11/1.36)**	**1.21 (1.06/1.38)**
Dietary calorie, kcal/day	+1SD	**1.20 (1.03/1.40)**	0.90 (0.73/1.12)
Dietary vitamin D, log IU/day	−1SD	**1.17 (1.01/1.36)**	1.09 (0.90/1.32)
Vitamin D supplement	No vs. yes	**8.81 (1.20/64.64)**	**24.32 (3.12/189.62)**

SD = standard deviation. eGFR = estimated glomerular filtration rate. **Bold character** for statistically significant odds ratio (95% CI not including one).

## Data Availability

The data underlying this article will be shared on reasonable request to the corresponding author. The data are stored in an institutional repository (https://repository.neuromed.it, accessed on 10 December 2021) and access is restricted by the ethical approvals and the legislation of the European Union. Supporting reported results can be found at Department of Epidemiology and Prevention, IRCCS Neuromed, 86077 Pozzilli, IS, Italy.

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
