# Peer review of "Correlates of Calcidiol Deficiency in Adults—Cross-Sectional, Observational, Population-Based Study"

_nutrients, 2022, doi:10.3390/nu14030459_

Round 1

Reviewer 1 Report

The study titled "Correlates of Calcifiol Deficiency in Adults - Cross-sectional, observation, population-based study" by Cirillo and colleagues investigates an important topic. The study seems well-designed, scientifically sound, and the manuscript could improve grammar in a couple of locations. With that said, the overall impact is uncertain. Clarifying the impact of this manuscript and impact on patients would be beneficial.

Author Response

The authors are grateful for the Reviewer’s comments and suggestions.

Comment

Response

Clarifying the impact of this manuscript and impact on patients would be beneficial.

Four lines were added in CONCLUSIONS to clarify the implications of the manuscript (lines 318-321)

Reviewer 2 Report

Thank you for your contribution.

I recomnmend to add the mean serum 25OHD in table 1, in men and women.

Add the more detail of reason why you pick the value of 12ng/ml for cut-off value.

Author Response

The authors are grateful for the Reviewer’s comments and suggestions.

Comment

Response

I recomnmend to add the mean serum 25OHD in table 1, in men and women.

The revised version includes mean±SD for men and women, separately. These additional data were included in the text of DESCRIPTIVE STATISTICS (lines 157-158). Gender-specific data on calcidiol were not included in Table 1 because the Table that does not report gender data for any variable.

Add the more detail of reason why you pick the value of 12ng/ml for cut-off value.

The revised version of INTRODUCTION gives details about the rationale of the 12 ng/mL threshold (lines 50-51)